# AdaDeDup: Adaptive Hybrid Data Pruning for Efficient Object Detection Training

## Abstract

The computational burden and inherent redundancy of large-scale datasets challenge the training of contemporary machine learning models. Data pruning offers a solution by selecting smaller, informative subsets, yet existing methods struggle: density-based approaches can be task-agnostic, while model-based techniques may introduce redundancy or prove computationally prohibitive. We introduce Adaptive De-Duplication (AdaDeDup), a novel hybrid framework that synergistically integrates density-based pruning with model-informed feedback in a cluster-adaptive manner. AdaDeDup first partitions data and applies an initial density-based pruning. It then employs a proxy model to evaluate the impact of this initial pruning within each cluster by comparing losses on kept versus pruned samples. This task-aware signal adaptively adjusts cluster-specific pruning thresholds, enabling more aggressive pruning in redundant clusters while preserving critical data in informative ones. Extensive experiments (>50k GPU hours) on large-scale object detection benchmarks (Waymo, COCO, nuScenes) using standard models (BEVFormer, Faster R-CNN) demonstrate AdaDeDup's advantages. It significantly outperforms prominent baselines, substantially reduces performance degradation (e.g., over 54% versus random sampling on Waymo), and achieves near-original model performance while pruning 20% of data, highlighting its efficacy in enhancing data efficiency for large-scale model training. Code is open-sourced[1].

## 1 Introduction

Training deep learning models in computer vision requires diverse datasets of influential points to achieve state-of-the-art performance, motivating extensive large-scale data collection efforts. However, data processing, labeling, and training compute costs may make training with a complete dataset impractical (Mahmood et al., 2022; Abbas et al., 2023; Mahmood et al., 2025). Moreover, large-scale datasets may have high degrees of data redundancy, imbalance, and label noise (Sorscher et al., 2022). Consequently, it is preferable in practice to train deep learning models with smaller subsets from a large collected dataset (Shen et al., 2025; Abbas et al., 2024).

*Data pruning* is the task of selecting a subset from a larger dataset such that training a model with this subset achieves comparable or even superior performance to training on the complete dataset. Data pruning methods fall into two main categories: density-based and model-based (Sorscher et al., 2022). Density-based methods leverage the underlying distribution of the dataset to identify representative subsets that approximate the original data distribution with fewer samples. Although computationally efficient, these methods may fail to capture nuanced task-specific relationships by discarding rare but influential data points (Abbas et al., 2023; Shen et al., 2025; Abbas et al., 2024; Griffin et al., 2024; Chai et al., 2023b; Xia et al., 2022). In contrast, model-based approaches assess the relevance of samples or subsets by measuring their impact on downstream task performance through training loss or gradients. However, these methods often incur substantial computational overhead from repeated model training or inference and may overly emphasize challenging samples, including noisy or mislabeled data points (Tan et al., 2023; Evans et al., 2024; Coleman et al., 2019; Lee et al., 2024; Chai et al., 2023a).

While significant advancements in data pruning have been made in natural language processing and image classification, the inherent challenges of data pruning are exacerbated for large-scale object

---

[1]Code repository: `https://anonymous.4open.science/r/AdaDeDup/`.

detection. Real-world detection datasets often exhibit less curation and significant data imbalance, offering a fertile ground for improving data efficiency. Yet, the scale of training tasks renders model-based pruning methods impractical. Although density-based methods could offer sufficient scalability, these pruning methods assume access to a high-quality pre-trained embedding space which does not hold for real-world detection applications such as autonomous driving (Shen et al., 2025). Images can be crowded with objects of significant heterogeneity (e.g., size, frequency, difficulty) (Caesar et al., 2020; Sun et al., 2020; Gupta et al., 2019), meaning visual similarity is insufficient and necessitating additional, task-specific designs. **Posing prominent gaps between existing data research and practical needs, these inherent complexities demand a tailored approach to data pruning for large-scale object detection training**.

In this paper, we develop **Ada**ptive **De**-**Dup**lication (ADADEDUP), a data-pruning framework for large-scale object detection datasets that combines the best of both density-based and model-based approaches. ADADEDUP distinguishes itself by proposing a novel *adaptive hybrid pruning* strategy. It initiates with efficient density-based pruning within semantic clusters (leveraging techniques akin to those in (Shen et al., 2025; Abbas et al., 2024)). Critically, it then introduces a model-informed adaptation step: feedback from a proxy model's loss, comparing initially kept versus pruned samples *within each cluster*, is used to dynamically adjust the pruning intensity for that specific cluster. This cluster-specific adaptation allows ADADEDUP to refine the initial density-based decisions in a targeted manner, recovering informative samples in certain regions while enabling more aggressive pruning where redundancy appears high according to model feedback. This synergistic approach provides a better balance between computational efficiency, task relevance, and performance preservation compared to existing purely density-based or model-based approaches, while maintaining simplicity for practical implementation.

In operation, given a large dataset, ADADEDUP first clusters the data using semantic features obtained from an off-the-shelf pretrained Vision-Language Model (VLM) that yields strong representations of an entire image and all objects within it. ADADEDUP then leverages model loss from a proxy model evaluated on samples within these clusters to adaptively adjust the intensity of each cluster while preserving samples in more informative ones. This process yields a computationally efficient algorithm that simultaneously captures the distribution of the large-scale dataset while pruning clusters that are least relevant to the downstream task. Finally, we demonstrate the effectiveness of ADADEDUP through extensive experiments on three large-scale, challenging detection benchmarks, i.e., Waymo (Sun et al., 2020), COCO (Lin et al., 2014), and nuScenes (Caesar et al., 2020). **This is the first work combining density-based data pruning with cluster-adaptive feedback for object detection.** Figure 1 visualizes the insights and motivations for developing ADADEDUP and Figure 2 demonstrates effectiveness on Waymo dataset.

Our key contributions are: (1) We propose ADADEDUP, a novel adaptive data pruning framework that adapts density-based and model-based criteria at a cluster-specific level. (2) We introduce a mechanism to estimate the impact of pruning within clusters using proxy model loss and adaptively adjusting cluster-specific pruning ratios. (3) We conduct comprehensive empirical evaluations demonstrating that ADADEDUP significantly outperforms strong baselines, including Random Downsampling, visual deduplication (CLIP-DeDup), and state-of-the-art semantic deduplication (VLM-SSE), across various datasets and pruning ratios. (4) We show that ADADEDUP substantially reduces the performance degradation associated with data pruning, achieving near-original performance at moderate pruning levels (e.g., 20% pruning on Waymo, 15% on COCO) and offering significant data efficiency improvements (e.g., matching Random Downsampling performance with 15-20% less data).

While the framework is general enough to be applied to other vision tasks, its primary innovation and advantage are most pronounced in domains like object detection where the gap between embedding similarity and true data value is largest. Results presented in the paper amounts to >50k GPU hours and the demonstrated effectiveness serves as the most direct evidence for the method's utility in its intended application.

## 2 RELATED WORK

Efficiently training models on large-scale datasets has driven extensive research into data pruning, which seeks to identify a representative subset of training data to minimize performance degradation compared to using the full dataset, while reducing computational load (Mahmood et al., 2022; 2025; Sorscher et al., 2022; Kang et al., 2023). This task is often framed as a bi-level optimization

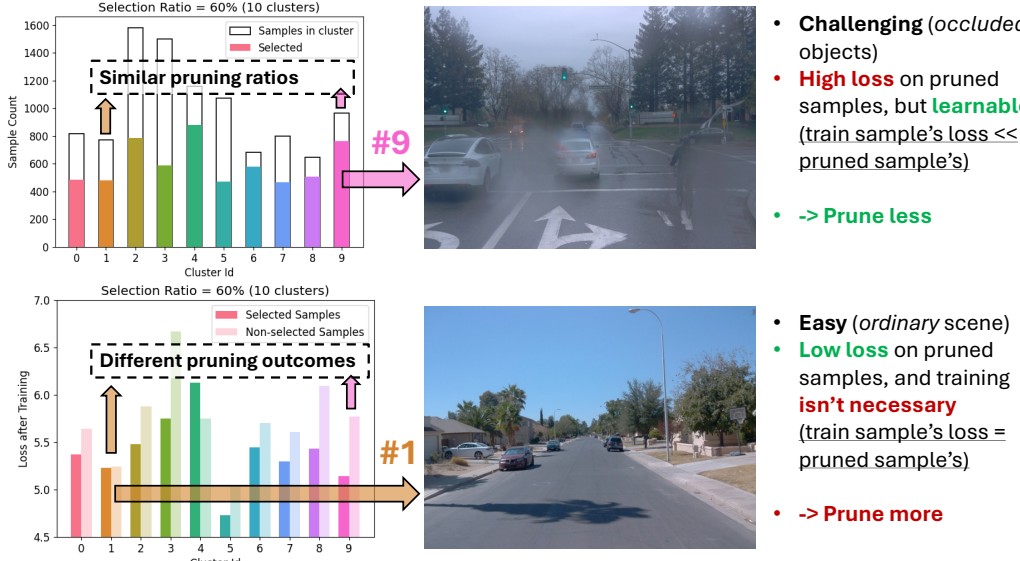

- **Challenging** (*occluded* objects)
- **High loss** on pruned samples, but **learnable** (train sample's loss << pruned sample's)
- **-> Prune less**

- **Easy** (*ordinary* scene)
- **Low loss** on pruned samples, and training **isn't necessary** (train sample's loss = pruned sample's)
- **-> Prune more**

Figure 1: Illustration on density-based data pruning (Shen et al., 2025) with 10 clusters on Waymo Dataset, retaining 60% of images. Samples in each clusters generally corresponding to different semantic/scenarios (e.g., rainy urban roads for cluster #9 (**Top Right**), clear residential neighborhood for cluster #1 (**Bottom Right**)). For a global uniform pruning threshold (e.g., removing the sample if there exist other samples within some threshold radius of $r$ on visual embeddings). Samples in different clusters are pruned with different ratios due to the different level of visual similarity (**Top Left**). Training a proxy model on the pruned data, **Bottom Left** shows the analysis of the trained model's loss on kept vs. pruned samples within different clusters. Clusters like #1 show pruned samples having lower/comparable loss to kept ones, suggesting redundancy. Conversely, clusters like #9 exhibit significantly higher loss on pruned samples, indicating that valuable, non-redundant information was discarded. This inherent heterogeneity in data motivates ADADEDUP's adaptive approach. Our work, ADADEDUP, addresses this by synergistically integrating density-based pruning with model-informed feedback at a cluster-adaptive level; it first performs an initial pruning and then uses signals from a proxy model to adaptively refine pruning intensity for each cluster.

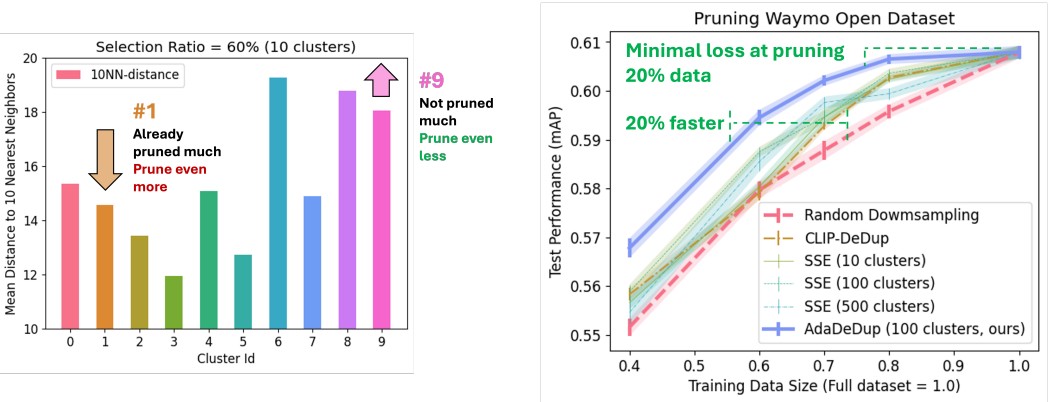

Figure 2: **Left:** Average distance to its 10 nearest neighbors in visual embeddings for samples in each cluster. A lower distance indicates higher redundancy where more samples will be removed during density-based data pruning. Insights from Figure 1 shows that despite cluster #1 having a higher proportion of samples pruned, it still exhibits higher redundancy. ADADEDUPwill prune more samples from clusters like #1 to make up data budget to add back samples from clusters like #9. **Right:** Performance (mAP ± std. dev.) of ADADEDUP compared to baselines on the Waymo dataset across various pruning ratios. By adaptively adjusting pruning intensity per cluster based on the type of insights shown left, ADADEDUP (blue line) consistently outperforms Random Downsampling, CLIP-DeDup, and VLM-SSE, especially at higher pruning ratios, demonstrating its effectiveness in preserving performance while maximizing data reduction.

problem and is NP-hard, necessitating heuristic solutions (Tan et al., 2023; Chai et al., 2023a). These heuristics primarily fall into density-based, model-based, and hybrid categories, each with distinct characteristics that our proposed method, ADADEDUP, aims to navigate.

**Density-Based Pruning**  One prominent line of work involves density-based data pruning, which leverages the common presence of redundant samples—such as exact duplicates, near-duplicates, or semantically similar instances—within machine learning datasets (Abbas et al., 2023; Tan et al., 2023; Coleman et al., 2019). These methods typically operate on sample representations in an embedding space, interpreting high local density as an indicator of redundancy (Sorscher et al., 2022; Shen et al., 2025; Abbas et al., 2024). Consequently, samples from dense clusters are pruned based on a predefined threshold, which reflects the proximity required for samples to be considered redundant (Abbas et al., 2023). While computationally appealing, the efficacy of density-based methods can be limited in complex tasks like object detection. In such scenarios, visual similarity in an embedding space does not always equate to semantic redundancy relevant to the task. For instance, visually similar images of a highway might represent a consistent scenario for autonomous driving, whereas subtle visual differences in construction site imagery could be critically important for model performance (Mahmood et al., 2022; Kang et al., 2023). Thus, relying solely on data distribution in an embedding space may lead to suboptimal pruning, potentially discarding task-relevant information.

**Model-Based Pruning**  Model-based selection techniques utilize feedback from a machine learning model to guide the data selection process. These can be broadly categorized:

- **Validation-Loss Modeling:** Some approaches aim to select data that maximizes model performance on a held-out validation set. They often estimate the influence of training points on validation loss, for example, by constructing counterfactual scenarios (Koh & Liang, 2017). While effective for optimizing validation accuracy, these methods might undervalue noisy or challenging samples whose impact on validation performance can be unpredictable.

- **Training-Loss Modeling and Coreset Selection:** Other methods prioritize samples by modeling their impact on the training process itself. This includes coreset selection methods, which aim to find a small weighted subset of data that approximates the loss or gradients of the full dataset (Sener & Savarese, 2017; Toneva et al., 2018; Paul et al., 2021; Borsos et al., 2020). GLISTER (Chai et al., 2023a) aim to select subsets whose gradient information closely matches the full dataset. Such techniques often emphasize samples that are challenging to fit, induce high uncertainty, or lie near decision boundaries (e.g., high-loss or high-gradient samples). Without careful regularization, these methods might risk over-prioritizing outliers or noisy samples.

A general challenge for many model-based approaches is their computational overhead, often requiring model training or extensive gradient computations. They can also often lead to selections lacking diversity if not explicitly managed, especially for large-scale datasets.

## 3  ADAPTIVE DATA PRUNING (ADADEDUP)

In this section, we first summarize the data pruning problem. We then propose our approach, ADADEDUP, which combines both density-based pruning with model-based feedback.

### 3.1  DATA PRUNING IN PRINCIPLE: BILEVEL OPTIMIZATION

In the context of an object detection problem, let the full dataset be $D_a = \{s_1, s_2, \ldots, s_n\}$, where each sample $s_i$ consists of an image paired with a set of bounding box-class labels. The total number of samples is $|D_a| = n$. To represent this selection, we define a set of binary variables $W_s = \{w_1, w_2, \ldots, w_n\}$, where $w_i \in \{0, 1\}$. A value of $w_i = 1$ indicates that sample $s_i$ is included in $D_s$, while $w_i = 0$ indicates its exclusion. Given a machine learning model parameterized by $\theta$ and a loss function $\ell(s; \theta)$ for an individual sample $s$, the model trained on the selected subset $D_s$ undergoes empirical risk minimization (ERM). The objective of ERM in this context is to find the parameters $\theta$ that minimize the weighted empirical risk: $L(\theta, W_s) = \sum_{i=1}^{n} w_i \ell(\theta; s_i)$ Standard ERM on the entire dataset $D_a$ is a special case where $w_i = 1$ for all $i \in \{1, \ldots, n\}$. In this scenario, the objective function simplifies to $L(\theta) = \sum_{i=1}^{n} \ell(\theta; s_i)$ (Borsos et al., 2020).

Data pruning seeks to identify a subset $D_s \subset D_a$ of size $|D_s| = m$, where $m < n$ is a predefined data budget. Consequently, the constraint $|D_s| = m$ is expressed as $\sum_{i=1}^{n} w_i = m$. A good pruned subset, characterized by a specific set of weights $\hat{W}_s = \{\hat{w}_1, \ldots, \hat{w}_n\}$, ensures that the weighted

loss $L(\theta, \hat{W}_s)$ serves as a reliable proxy for the full dataset loss $L(\theta)$. That is, optimizing $\theta$ based on $L(\theta, \hat{W}_s)$ should yield solutions that perform well when evaluated using $L(\theta)$. In this spirit, a natural goal is to determine a set of weights $\hat{W}_s$ such that we minimize the overall objective formulated as:

$$\min_{W_s} \mathcal{L}(\theta^*(W_s)) = \sum_{i=1}^{n} \ell(\theta^*(W_s); s_i) := \mathcal{J}(W_s), \text{ where } \theta^*(W_s) := \arg\min_{\theta} \mathcal{L}(\theta, W_s). \quad (1)$$

where the model parameterized by $\theta$ is trained with ERM on samples selected by $W_s$.

**The Challenge of Optimal Selection Necessitates Approximate Solutions**    The optimal selection of data subsets for training is a fundamentally challenging task, often formulated as a bi-level optimization problem (as detailed in Eq. equation 1). This problem is NP-hard, meaning exact solutions are generally intractable for datasets of non-trivial size. While general methods exist for solving bi-level optimization problems, such as those that compute gradients with respect to individual data point selection weights using explicit or implicit differentiation techniques, these approaches are typically computationally prohibitive, especially at the scale required for training object detection models (Chen et al., 2022; Bard, 1998). Existing work tries to approach it through heuristic methods under density-based data pruning or model-based data selection, but none of them is satisfactorily feasible/applicable/efficient in the context of object detection. *Density-based data pruning* (Sorscher et al., 2022; Shen et al., 2025; Abbas et al., 2024) leverages the prevalence of redundant samples (exact, near, or semantic duplicates (Abbas et al., 2023; Tan et al., 2023; Coleman et al., 2019)) by removing samples from high local density regions in an embedding space, guided by a density threshold (Abbas et al., 2023). *However, for object detection, visual similarity may not equate to semantic importance*—e.g., consistent highway scenes for autonomous vehicles versus nuanced, critical differences in construction sites (Mahmood et al., 2022; Kang et al., 2023)—rendering such methods potentially inefficient or inapplicable. Alternatively, *model-based data selection*, including coreset selection and active learning, utilizes model predictions. These are broadly categorized into training loss-based approaches (Sener & Savarese, 2017; Toneva et al., 2018; Paul et al., 2021) and validation loss-based approaches (Koh & Liang, 2017). Model-based methods can *risk overvaluing noise or mislabeled data, may lead to imbalanced selections lacking diversity, and often incur high computational costs, limiting their use in large-scale practical scenarios.*

### 3.2    Adaptive Density-based Pruning Strategies

**Limitations of Fixed Density-based Pruning.** Standard density-based pruning methods, relying on embedding space distances, often fall short in capturing complex image similarities robustly across varying data densities or semantic regions (Abbas et al., 2024; Evans et al., 2024). Such methods employ fixed, often global, thresholds, which can lead to either excessive information loss through aggressive pruning or insufficient redundancy reduction with conservative settings. This inflexibility highlights the challenge: density metrics alone serve as poor heuristics for the complex bilevel optimization underlying optimal data selection. To enhance data efficiency, particularly in nuanced tasks like object detection, a more granular and adaptive control over density-based pruning is essential. While some approaches adjust pruning strength based on inter-cluster geometry (Abbas et al., 2023), they remain agnostic to the downstream task, failing to adequately balance redundancy removal with the preservation of task-relevant information.

**Approaching Adaptive Pruning Parameter Optimization.**    For object detection, images within large datasets are often heterogeneous. Different regions or clusters of data may exhibit varying levels of redundancy and information value. A uniform pruning strategy applied globally is unlikely to be optimal. To render the bilevel optimization problem of data selection more tractable while retaining the benefits of density-based approaches, we propose an adaptive strategy. By integrating density-based pruning with model-based feedback and applying this hybrid approach adaptively at a cluster level, ADADEDUP aims to adapt the pruning intensity specifically for each cluster, allowing for aggressive pruning where redundancy is high and conservative pruning where information content is critical, thereby achieving a better balance between data reduction and performance preservation. Our high-level approach focuses on three core ideas: (1) parameterizing the pruning policy to narrow the decision space, (2) employing a tractable zero-order estimation for the policy gradient, and (3) efficient one-shot policy update driven by empirical insights.

1. **Parameterizing the pruning policy to narrow the decision space.** We narrow the decision space from assigning individual weights to each data point to optimizing a set of parameters $\lambda$ that govern the pruning policy. Inspired by research indicating that model performance can be

modeled as a function of data selection (Ilyas et al., 2022) and the feasibility of gradient-based optimization for data selection (Kang et al., 2023; 2024b), we aim to optimize the parameters $\lambda$ of a density-based selection policy $f$. Given a full dataset $D_a$, the selected subset $W_s$ is determined by $W_s = f(D_a, \lambda)$. Our objective is to optimize $\lambda$ to minimize a performance metric $\mathcal{J}(W_s)$. The gradient of this objective with respect to $\lambda$ can be expressed using the chain rule: $\frac{\partial \mathcal{J}(W_s)}{\partial \lambda} = \frac{\partial \mathcal{J}(W_s)}{\partial W_s} \cdot \frac{\partial W_s}{\partial \lambda}$. Here, $\frac{\partial \mathcal{J}(W_s)}{\partial W_s}$ represents the marginal contribution of samples in $W_s$ to the objective (e.g., total loss of a model trained on $W_s$, evaluated on $D_a$), and $\frac{\partial W_s}{\partial \lambda}$ reflects how changes in $\lambda$ affect the composition of $W_s$. Direct optimization of $\mathcal{J}(W_s)$ through this gradient is often prohibitively expensive due to the need for repeated model re-training to evaluate $\mathcal{J}(W_s)$ and its derivatives accurately (Borsos et al., 2020). The term $\frac{\partial \mathcal{J}(W_s)}{\partial (W_s)_p}$ for an unselected (pruned) sample $s_p$ (where $(W_s)_p = 0$) estimates the counterfactual impact on $\mathcal{J}(W_s)$ had $s_p$ been included in $W_s$.

2. **Zero-order estimation for the policy gradient.** Parameterizing the pruning policy to narrow the decision space. To make the estimation of $\frac{\partial \mathcal{J}(W_s)}{\partial W_s}$ tractable, we employ a zero-order approximation based on local cluster characteristics. For a pruned sample $s_p$ from a cluster $c$, and a kept sample $s_k$ from the same cluster $c$ that is close to $s_p$ in the embedding space, we approximate the potential impact of including $s_p$. This is achieved by comparing the loss of the current model $\theta^*(W_s)$ (trained on the currently selected subset $W_s$) on $s_p$ with its loss on $s_k$. The difference, $\ell(\theta^*(W_s), s_p) - \ell(\theta^*(W_s), s_k)$, serves as a proxy for the marginal contribution of $s_p$. A small or negative difference suggests $s_p$ might be redundant, as the model generalizes well to it from $s_k$. Conversely, a large positive difference indicates that $s_p$ contains information not captured by $s_k$ and other kept neighbors, implying information loss due to its pruning. This estimated loss difference approximates $\frac{\partial \mathcal{J}(W_s)}{\partial (W_s)_p}$. If $\lambda_c$ is a parameter controlling pruning for cluster $c$ (e.g., a local density threshold), the term $\frac{\partial W_s}{\partial \lambda_c}$ in Eq. equation 1 describes how adjusting $\lambda_c$ alters sample selection within $c$. The overall gradient $\frac{\partial \mathcal{J}(W_s)}{\partial \lambda_c}$ thus guides the adjustment of $\lambda_c$.

3. **Efficient one-shot policy update.** Armed with these tractable estimations, we can devise update strategies for $\lambda$. While iterative optimization of $\lambda$ can still be computationally intensive, we observe empirically that the optimization landscape for data pruning in tasks like object detection can be amenable to simpler, effective solutions. The sign and relative magnitude of the approximated marginal contributions (the local loss differences) often provide a sufficiently robust signal. This allows for efficient, targeted adjustments to the pruning policy parameters $\lambda_c$ for different clusters, aiming to reduce pruning in clusters with high estimated information loss and increase pruning where redundancy is still high with minimal information content. In practice, we find that a single update step, adjusting the selection status for a small percentage (e.g., 5-10%) of samples based on these signals yields substantial improvements often comparable to complex iterative optimization or line search procedures.

### 3.3 Proposed Method: Adaptive Data Pruning (AdaDeDup)

Our proposed method, Adaptive Data Pruning (AdaDeDup), operationalizes principles outlined in Section 3.2 by implementing a two-stage process synergizing density-based selection with model-informed feedback to adaptively adjust pruning at the cluster level. AdaDeDup requires the initial dataset $D_a$, a target subset size $m$ (or equivalently, a pruning ratio $\gamma = (n - m)/n$), an proxy model $\tilde{\mathcal{A}}$, and its associated loss function $\mathcal{L}$. The proxy model can be the target model intended for final training or a computationally cheaper alternative to guide the pruning process (Coleman et al., 2019). *The detailed steps for operation are presented in Algorithm 1 in Appendix. D.*

#### 3.3.1 Stage 1: Initial Clustered Density-Based Pruning

The first stage establishes a baseline pruned dataset.

1. **Feature Extraction and Clustering:** Samples in $D_a$ are transformed into semantic feature vectors (e.g., using embeddings from pre-trained vision models or Vision-Language Models (VLMs) suitable for object detection). These features are then used to group the $n$ samples into $K$ distinct clusters, $C = \{c_1, \ldots, c_K\}$, using a standard clustering algorithm (e.g., $K$-means). This step groups semantically or visually similar items, forming the basis for cluster-specific adaptation.

2. **Initial Pruning:** An initial density-based pruning is applied globally or within clusters. For instance, a global distance threshold $\tau$ can be used for de-duplication, or prototypes can be

selected from each cluster. This results in an initial selected subset $D_s^{(0)}$ containing exactly $m$ samples, and an initially pruned set $D_p^{(0)} = D_a \setminus D_s^{(0)}$. This step implicitly defines initial per-cluster pruning ratios $\gamma_i = |D_p^{(0)} \cap c_i|/|c_i|$ for each cluster $c_i$.

### 3.3.2 STAGE 2: MODEL-INFORMED ADAPTIVE RE-PRUNING

This stage refines the initial pruning by leveraging feedback from the proxy model $\tilde{\mathcal{A}}$ to assess the utility of data within each cluster, implementing the zero-order gradient estimation technique.

1. **Evaluating Pruning Impact per Cluster:** The proxy model $\tilde{\mathcal{A}}$ is trained on the initially selected subset $D_s^{(0)}$, yielding model parameters $\theta^*(D_s^{(0)})$. This trained model, denoted $\tilde{\mathcal{A}}(D_s^{(0)})$, is then used to evaluate the loss $\mathcal{L}$ for all samples in the original dataset $D_a$. For each cluster $c_i \in C$, we aggregate the losses on its initially selected samples and its initially pruned samples:

$$\ell_i^s := \sum_{s \in D_s^{(0)} \cap c_i} \mathcal{L}(\tilde{\mathcal{A}}(D_s^{(0)}), s), \quad \text{and} \quad \ell_i^p := \sum_{s \in D_p^{(0)} \cap c_i} \mathcal{L}(\tilde{\mathcal{A}}(D_s^{(0)}), s). \tag{2}$$

We then compute a differential loss signal for cluster $c_i$: $\Delta\ell_i = \ell_i^s - \ell_i^p$. This $\Delta\ell_i$ serves as a proxy indicating the relative information content or difficulty of the kept versus pruned portions of cluster $c_i$, guiding the adjustment of its pruning parameter $\lambda_c$ (represented by $\gamma_i'$):

   - If $\Delta\ell_i > 0$ (i.e., $\ell_i^s > \ell_i^p$): The samples initially kept in $c_i$ are, on average, "harder" (exhibit higher loss) for the proxy model than those initially pruned from $c_i$. This suggests that the pruned samples from this cluster were relatively "easier" or more redundant with respect to the information captured by $\tilde{\mathcal{A}}(D_s^{(0)})$. ADADEDUP interprets this as an opportunity to prune *more* aggressively from cluster $c_i$.
   - If $\Delta\ell_i < 0$ (i.e., $\ell_i^s < \ell_i^p$): The samples initially kept in $c_i$ are "easier" than those pruned. This implies that potentially more informative or challenging samples were discarded from $c_i$ during the initial pruning. ADADEDUP then aims to prune *less* aggressively from this cluster to recover or retain these samples.

2. **Adaptive Re-pruning Strategy:** The differential losses $\Delta\ell_i$ guide the adjustment of pruning strength for each cluster. We define a scaled signal $\tilde{\Delta}\ell_i = \alpha_i \Delta\ell_i$. Consistent with Algorithm 1, $\alpha_i$ is $\alpha_+$ if $\Delta\ell_i > 0$ and $\alpha_-$ otherwise, where $\alpha_+$ and $\alpha_-$ are *positive* scaling constants. The target adjusted pruning ratio $\gamma_i'$ for cluster $c_i$ is then calculated as: $\gamma_i' := \text{clip}(\gamma_i + \beta \cdot \tilde{\Delta}\ell_i, 0, 1)$, where $\beta > 0$ is an adaptation strength hyperparameter. With positive $\alpha_+, \alpha_-$ in the update:

   - If $\Delta\ell_i > 0$, then $\tilde{\Delta}\ell_i = \alpha_+ \Delta\ell_i > 0$. The term $\beta \cdot \tilde{\Delta}\ell_i$ is positive, thus $\gamma_i'$ tends to increase, leading to *more* pruning in cluster $c_i$.
   - If $\Delta\ell_i < 0$, then $\tilde{\Delta}\ell_i = \alpha_- \Delta\ell_i < 0$. The term $\beta \cdot \tilde{\Delta}\ell_i$ is negative, thus $\gamma_i'$ tends to decrease, leading to *less* pruning in cluster $c_i$.

   To ensure the overall target number of samples $m$ is preserved (i.e., $\sum_{i=1}^K |c_i|(1 - \gamma_i') = m$), the $\tilde{\Delta}\ell_i$ values are typically normalized before computing the final $\gamma_i'$. Further, instead of conducting a line search, selecting $\beta$ such that update on pruning policy affects $5 - 10\%$ of sample selection leads to satisfactory empirically results in object detection tasks.

3. **Final Data Selection:** Finally, the dataset is re-pruned from the original $D_a$. For each cluster $c_i$, a new cluster-specific density threshold $\tau_i$ is determined and applied such that it results in keeping $|c_i|(1 - \gamma_i')$ samples from $c_i$. The final selected dataset $D_s$ is the union of these newly selected samples from all clusters. This one-shot adjustment, guided by empirical model-based signals, implements the efficient policy update strategy.

## 4 EMPIRICAL RESULTS

### 4.1 EXPERIMENT SETUP

**Datasets and Models:** We evaluate ADADEDUP on three standard object detection benchmarks: **Waymo Open Dataset (Waymo)** (Sun et al., 2020): A large-scale autonomous driving dataset. We use a 2Hz subsampled version from Li et al. (2024), focusing on vehicle, pedestrian, and cyclist detection. **COCO (Common Objects in Context) 2017** (Lin et al., 2014): A widely-used vision benchmark with 118k training images spanning 80 object categories. **nuScenes Dataset** (Caesar et al.,

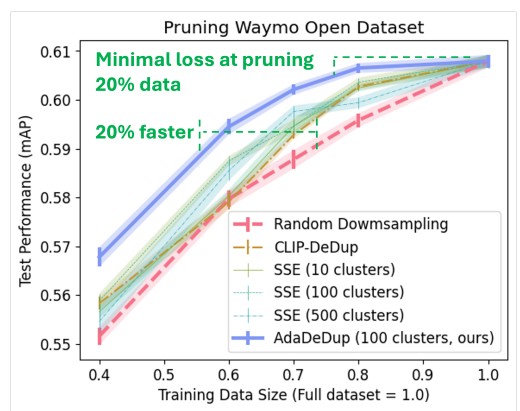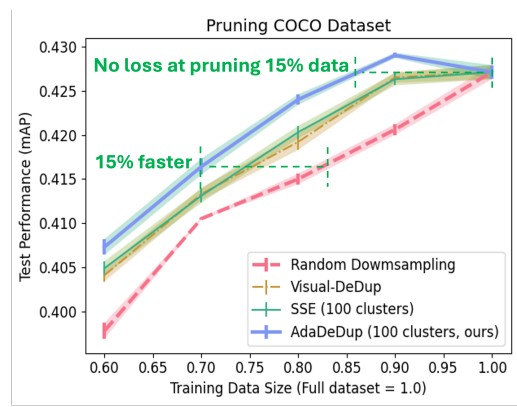

Figure 3: mAP ($\pm$ std. dev.) vs. data retained on Waymo. ADADEDUP shows significant performance retention.

Figure 4: mAP ($\pm$ std. dev.) vs. data retained on COCO. ADADEDUP consistently leads.

2020): An autonomous driving dataset with 1,000 scenes (approx. 28k training images) annotated at 2Hz across 10 object classes. For Waymo and nuScenes, we use BEVFormer-S (a static variant of BEVFormer (Li et al., 2024)) with a ResNet101-DCN (He et al., 2016) backbone. For COCO, we use Faster R-CNN (Ren et al., 2016) with a ResNet-101-FPN backbone via Detectron2 (Wu et al., 2019). All models are trained for the same number of epochs on full and pruned datasets, maintaining the original learning rate schedules. All model training was conducted on single nodes with 8x NVIDIA V100 GPU. Each training run took around one day. Each experiment has been repeated over 3 runs with standard deviations reported alongside main results. VLM captioning was conducted on a single NVIDIA A5880 Ada GPU. Further details on dataset preprocessing, model configurations, and training hyperparameters are provided in Appendix E. These datasets and applications are inherently characterized by the very challenges that motivated this work (diverse scenarios, vast differences in object frequency/size/occlusion). Experiments conducted in this work spent **>50k GPU hours**. **Baselines:** We compare ADADEDUP against three representative baselines: **Random Downsampling (Random)**: Uniformly samples a subset without replacement. **Visual Deduplication (CLIP-DeDup)**: A density-based method that discards samples if their visual embeddings are within a threshold. We use CLIP-ViT-L/14 (Radford et al., 2021) for Waymo/nuScenes and GroundingDINO (Liu et al., 2024b) for COCO. For Waymo/nuScenes, deduplication on the front-view image leads to discarding all images in the corresponding multi-view scene. **VLM-SSE (SSE)** (Shen et al., 2025): State-of-the-art semantic deduplication. It generates image captions using a VLM (LLaVA-1.5-13B (Liu et al., 2024a)) to form semantic clusters, then performs visual deduplication within each cluster using a uniform threshold. We test with 10, 100, and 500 clusters.

## 4.2 RESULTS ON PRUNING WAYMO OPEN DATASET

We pruned Waymo to $80\%$, $70\%$, $60\%$, and $40\%$ of its original size. For ADADEDUP, proxy models were trained on $\leq 10k$ samples. Results (mean average precision, mAP $\pm$ std. dev. over 3 runs) are in Figure 3 (left). Additional results and visualizations are available in Appendix E. ADADEDUP consistently outperforms all baselines across all pruning ratios. Notably, at $20\%$ pruning ($80\%$ data retained), ADADEDUP achieves nearly identical performance to training on the full dataset, offering a direct $20\%$ data efficiency gain with minimal performance degradation. Furthermore, ADADEDUP using $60\%$ of the data performs comparably to Random Downsampling using $80\%$ of the data. *For up to $40\%$ pruning, ADADEDUP reduces performance loss by at least $54\%$ compared to Random Downsampling and at least $35\%$ compared to other baselines.* Among baselines, VLM-SSE (100 clusters) generally performed best, particularly at moderate pruning ratios ($40\%$). CLIP-DeDup was effective at smaller pruning ratios but its performance degraded more rapidly than VLM-SSE. Random Downsampling consistently performed the worst.

## 4.3 RESULTS ON PRUNING COCO DATASET

We pruned COCO to $90\%$, $80\%$, $70\%$, and $60\%$ of its original size. ADADEDUP's proxy models were trained on $\leq 30k$ samples ($\sim 25\%$ of COCO train). Results (mAP $\pm$ std. dev. over 3 runs) are shown in Figure 4 (right). Again, ADADEDUP significantly outperforms all baselines. At $10\%$ pruning ($90\%$ data retained), ADADEDUP shows negligible performance loss, effectively improving data

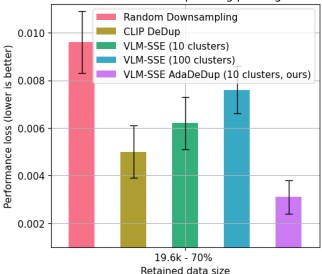

| nuScenes/mAP Retained Size | Random Downsampling | CLIP DeDup | VLM-SSE (10 clusters) | VLM-SSE (100 clusters) | ADADEDUP (ours) |
|---|---|---|---|---|---|
| **Full (28k - 100%)** | | | $0.3759_{\pm 0.0010}$ | | |
| **19.6k - 70%** *(Perf. Drop)* | $0.3663_{\pm 0.0013}$ *(-0.0096)* | $0.3709_{\pm 0.0011}$ *(-0.0050)* | $0.3697_{\pm 0.0011}$ *(-0.0062)* | $0.3683_{\pm 0.0010}$ *(-0.0076)* | $\mathbf{0.3728}_{\pm 0.0007}$ ***(-0.0031)*** |

Figure 5: Performance on nuScenes when pruned to $70\%$ data. **Left:** Table showing mAP ($\pm$ std. dev.) and performance drop from full dataset. **Right:** Visualization of performance drop (mAP $\pm$ std. dev.). ADADEDUP demonstrates the smallest performance degradation.

efficiency by $10\%$. Models trained on $70 - 80\%$ of data pruned by ADADEDUP perform comparably to models trained on $90\%$ of data from Random Downsampling, a $\sim 15 - 20\%$ relative data reduction for similar performance. *For up to $20\%$ pruning,* ADADEDUP *reduces performance loss by at least $66\%$ compared to Random Downsampling and at least $40\%$ compared to other baselines.*

### 4.4 RESULTS ON PRUNING NUSCENES DATASET

We evaluated pruning nuScenes to $70\%$ of its original size ($30\%$ pruning). ADADEDUP's proxy models used $\leq 10k$ samples. Results are presented in Figure 5. At this $30\%$ pruning ratio, ADAD-EDUP outperforms all baselines, *reducing performance loss by $68\%$ compared to Random Downsampling and at least $38\%$ compared to other baselines.* CLIP-DeDup was the second-best performer, surpassing VLM-SSE variants on this dataset at this specific pruning level. VLM-SSE with 10 clusters performed slightly better than with 100 clusters.

**Remark.** *Existing density-based pruning methods often require training proxy models for feature embedding (Abbas et al., 2023) or discovering "prototypical samples" (Sorscher et al., 2022), which is typically considered as a standard step. Seminal work Coleman et al. (2019) formally discussed selecting data based on proxy models, which is introduced efficient approach compared to selection based on the full model.* ADADEDUP *does not require proxy models stronger than those and remains dramatically cheaper than model-based pruning methods. Further, this work is designed for the* ***one-round data pruning*** *problem and conduct dataset reduction using only proxy models. In this practical setup, data pruning is separated from actual model training, allowing distributing workload and better quality control during the model development cycle. Dynamic data sampling or curriculum learning methods, which improve efficiency during actual training on entire original dataset, cater to different downstream scenarios and the results are not directly comparable. We defer the full discussion to Appendix B.*

## 5 CONCLUSION

This paper introduced ADADEDUP, a novel hybrid data pruning framework that synergistically integrates density-based clustering with model-informed, cluster-adaptive feedback to enhance data efficiency in training large-scale models, particularly for object detection. ADADEDUP first establishes an initial pruned set via density analysis and then utilizes a proxy model to assess information content within each cluster, adaptively refining pruning thresholds to preserve informative data while aggressively pruning redundant regions. Experiments on major object detection benchmarks, spending $>50k$ GPU hours, demonstrated that ADADEDUP significantly outperforms prominent baselines, achieving near-original model performance with substantial data reductions (e.g., up to 20%) and showcasing considerable improvements in data efficiency. These findings underscore the value of adaptive, hybrid strategies in data pruning for object detection training at scale. **Limitations and Future Work:** This work does not explicitly explore the impact on bias mitigation. Fairness improvements and data pruning are generally orthogonal effort, typically developed in separate works and improved iteratively. For AdaDedup, fairness can be an additional objective to be incorporated into the adaptive adjustment (Slyman et al., 2024). Besides, it is a often a challenge for data pruning methods distinguish the valuable "hard" samples and outliers (e.g., mislabeled samples). This work, similar to current density-based data pruning methods, caters to the general data curation/dataset reduction scenarios. On orthogonal research lines, data influence (e.g., Koh & Liang (2017)) and data valuation (e.g., Just et al. (2023)) methods specialize at these diagnostic use cases. Applying data filtering techniques in conjecture with data pruning is promising path for future exploration. These integration could further strengthen the benefits of adaptive data pruning.

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

# Appendices

## A  BROADER IMPACT STATEMENT

ADADEDUP aims to improve the efficiency of training large-scale machine learning models, offering several positive broader impacts. By reducing data and computational requirements, it can enhance the accessibility of advanced AI, foster environmental sustainability through lower energy consumption ("Green AI"), and accelerate innovation in applied domains reliant on large datasets. Further, a critical consideration is that data pruning, including ADADEDUP, holds the potential to reduce existing dataset biases if carefully managed. If certain demographic groups or rare but critical scenarios are sufficiently represented in evaluation, they might have a higher chance to be perserved. Therefore, practitioners must conduct thorough fairness audits and evaluate model performance across diverse subpopulations. We advocate for responsible application of such techniques and highlight the integration of fairness metrics directly into the pruning process as an important avenue for future work, ensuring efficiency gains do not compromise equity.

## B  DISCUSSIONS: PROXY MODEL, EFFICIENCY, ONE-ROUND DATA PRUNING

**Proxy Model, Efficiency, and Object Detection Training.** Seminal work Coleman et al. (2019) formally discussed selecting data based on proxy models, which is introduced efficient approach compared to selection based on the full model. Existing density-based pruning methods also require training proxy models for feature embedding (Abbas et al., 2023) or discovering "prototypical samples" (Sorscher et al., 2022), which is typically considered as a standard step. ADADEDUP *does not require proxy models stronger than those and remains dramatically cheaper than model-based pruning methods*, which are often formulated as bi-level optimization with integer constraints that may not be solvable in a reasonable time (Borsos et al., 2020). Besides, the loss evaluation step unique to ADADEDUP is fairly cheap. Different from model-based methods often requiring evaluating the loss of every sample, ADADEDUP only evaluates the loss at the cluster level, which can be estimated efficiently on a subset of samples for each cluster. This only needs a single forward pass for the proxy model on subsampled data. Compared to the original model trained on full dataset for up to $24 \sim 48$ epochs, the computation overhead for loss evaluation in ADADEDUP is mostly negligible.

**One-round Data Pruning.** Besides, for large-scale model development, data curation and model training are often conducted with different teams working in parallel. This work is designed for the **one-round data pruning** problem improving over density-based data pruning methods and conducting dataset reduction using only proxy models. In this practical setup, data pruning is separated from actual model training, allowing distributing workload and better quality control during the model development cycle. In comparison, dynamic data sampling (Qin et al., 2023) or curriculum learning methods (Soviany et al., 2022), which improve efficiency during actual training on entire original dataset, cater to different downstream scenarios than one-round data pruning methods. Empirical evaluation is conducted different setups and the results are not directly comparable.

## C  EXTENDED RELATED WORK

The challenge of efficiently training models on large-scale datasets has spurred significant research into methods for reducing data volume while preserving essential information. Techniques include data pruning (concept overlaps with data selection) (Mahmood et al., 2022; 2025; Sorscher et al., 2022; Kang et al., 2023), dataset distillation (Sachdeva & McAuley, 2023), and coreset selection (Lee et al., 2024; Killamsetty et al., 2021b;a). **Data pruning**, including this work, aims to select a subset of the original training data that minimizes performance loss compared to training on the full data. In essence, this could be viewed as an optimization problem over the variable of training data selection to maximize final model performance. Evaluation model performance requires training the model, which is a costly optimization problem by itself. This renders data pruning a bi-level optimization problem (Mahmood et al., 2025), and selecting an optimal subset is NP-hard, necessitating heuristic approaches (Tan et al., 2023; Chai et al., 2023a). These approaches often fall into data-centric (density-based) or model-centric (model-based) categories.

**Density-Based Methods** (or data-centric methods) operate directly on the data distribution, typically in a feature space, without necessarily requiring model training during selection. They primarily aim to reduce redundancy by identifying and removing samples considered similar to others. Density-based

approaches have been championed following the seminal work (Sorscher et al., 2022), which suggests samples near cluster centers are "easier" and pruning them is less harmful, yielding a straightforward approach achieving satisfactory performance comparable to most performant approaches with an intensive computational overhead. Common strategies include prototype selection and distance-based thresholding, where one sample from a pair within a certain distance is discarded (Sorscher et al., 2022). For example, SemDeDup (Abbas et al., 2023) argues samples close in an embedding space are redundant and pruning them minimally impacts performance. A key challenge for these methods lies in the implicit assumption of an "ideal" embedding space where distances directly indicate usefulness globally, irrespective of the task or model. Recent efforts aim to solve this issue. SSE (Shen et al., 2025) uses Vision-Language Models (VLMs) to generate captions, enabling semantic clustering before applying visual deduplication within clusters, thus avoiding pruning visually similar but semantically different samples. However, the pruning criteria within clusters often remain uniform. Abbas et al. (2024) propose varying pruning criteria based on cluster geometry (e.g., density, distance to other clusters), arguing denser clusters can be pruned more aggressively. However, these assume that the geometric statistics in the embedding space fully capture sample relationships that are relevant to the considered task. Despite being computationally cheaper, density-based methods risk being task-agnostic since their pruning strategy does not consider the task at hand, and they may discard informative outliers or boundary samples. To address these limitations, we propose a pruning strategy that considers the performance loss of a trained model.

**Model-Based Methods** leverage information from a machine learning model to guide selection, aiming to retain samples deemed "informative" or "important" for learning. **Loss-based** methods prioritize high-loss samples, assuming they are "hard" or informative (Toneva et al., 2018; Paul et al., 2021; Tan et al., 2023). However, this can be sensitive to noise/mislabels and may select many similar hard examples, potentially harming diversity (Just et al., 2023). **Gradient-based** methods like GRAD-MATCH (Killamsetty et al., 2021a) or GLISTER (Killamsetty et al., 2021b) aim to select subsets whose gradient information closely matches the full dataset, minimizing gradient disparity or maximizing likelihood proxies. Influence functions also estimate sample importance based on their impact on model parameters or predictions (Koh & Liang, 2017; Pruthi et al., 2020). **Uncertainty-based** methods, which are common in active learning, select samples where the model is least confident, assuming high information gain (Evans et al., 2024). While task-relevant, model-based methods often incur significant computational overhead due to repeated model training, inference, or gradient computations (Coleman et al., 2019). Furthermore, selecting samples based purely on individual scores (like loss) can lead to homogeneous subsets lacking diversity—e.g., similar samples receiving similar scores (Kang et al., 2024a).

ADADEDUP distinguishes itself by proposing a novel *adaptive hybrid pruning* strategy specifically designed to bridge the gap identified above. It starts with efficient density-based pruning within semantic clusters (pluggable to density-based methods such as [cite]). Critically, it then introduces a model-informed adaptation step: feedback from a proxy model's loss, comparing initially kept versus pruned samples *within each cluster*, is used to dynamically adjust the pruning intensity (effective threshold) for that specific cluster. Unlike static hybrid methods (e.g., weighted combination of different scores) or globally applied adaptive criteria, this cluster-specific adaptation allows ADADEDUP to refine the initial density-based decisions in a targeted manner, recovering informative samples in regions while pruning more aggressively where redundancy appears high according to model feedback. Together, this provides a better balance between efficiency, task relevance, and performance preservation compared to existing approaches while preserving simplicity for easy implementation.

# D  METHOD AND ALGORITHM PROCEDURES

Detailed steps for implementing the two-stage process of ADADEDUP are presented in Algorithm 1. Code is open-sourced at `https://anonymous.4open.science/r/AdaDeDup/`.

**Algorithm 1** Adaptive Data Pruning (ADADEDUP)

---

**Input:** Initial dataset $D_a = \{s_1, \ldots, s_n\}$, target selected subset size $m$, proxy model $\tilde{\mathcal{A}}$, loss function $\mathcal{L}$, number of clusters $K$, adaptation strength $\beta$, positive scaling factors $\alpha_+, \alpha_-$.

**Output:** Final selected dataset $D_s$ with $|D_s| = m$.

1: Extract features $\{f_1, \ldots, f_n\}$ for each $s_i \in D_a$.
2: Cluster dataset $D_a$ into $K$ clusters, $C = \{c_1, \ldots, c_K\}$, based on features.
3: Perform initial global or per-cluster density-based pruning on $D_a$ to obtain $D_s^{(0)}$ with $|D_s^{(0)}| = m$, and $D_p^{(0)} = D_a \setminus D_s^{(0)}$.
4: **for** $i = 1, \ldots, K$ **do**
5: $\quad$ Compute initial pruning ratio for cluster $c_i$: $\gamma_i = \frac{|D_p^{(0)} \cap c_i|}{|c_i|}$.
6: Train proxy model $\tilde{\mathcal{A}}$ on $D_s^{(0)}$, obtaining $\tilde{\mathcal{A}}(D_s^{(0)})$.
7: **for** $i = 1, \ldots, K$ **do**
8: $\quad$ Calculate aggregated losses for samples in cluster $c_i$:

$$\ell_i^s = \sum_{s \in D_s^{(0)} \cap c_i} \mathcal{L}(\tilde{\mathcal{A}}(D_s^{(0)}), s), \quad \text{and} \quad \ell_i^p = \sum_{s \in D_p^{(0)} \cap c_i} \mathcal{L}(\tilde{\mathcal{A}}(D_s^{(0)}), s).$$

9: $\quad$ Compute differential loss for cluster $c_i$: $\Delta\ell_i = \ell_i^s - \ell_i^p$.
10: Initialize list of *scaled differential losses (SDL)*.
11: **for** $i = 1, \ldots, K$ **do**
12: $\quad$ Set scaling factor: $\alpha_i = \alpha_+$ if $\Delta\ell_i > 0$, else $\alpha_i = \alpha_-$.
13: $\quad$ Compute scaled differential loss: $\tilde{\Delta\ell_i} = \alpha_i \cdot \Delta\ell_i$. Add to *SDL*.
14: Normalize $\tilde{\Delta\ell_i}$ values in *SDL* such that the constraint $\sum_{i=1}^{K} |c_i|(\gamma_i + \beta \cdot \tilde{\Delta\ell}_{i_{\text{norm/adj}}}) = n - m$ (target total pruned count) is met, while respecting $0 \leq \gamma_i' \leq 1$. Let the adjusted values be $\tilde{\Delta\ell_i}^*$.
15: Initialize final selected set $D_s \leftarrow \emptyset$.
16: **for** $i = 1, \ldots, K$ **do**
17: $\quad$ Compute adjusted cluster pruning ratio: $\gamma_i' = \gamma_i + \beta \cdot \tilde{\Delta\ell_i}^*[i]$.
18: $\quad$ Clip $\gamma_i'$: $\gamma_i' = \min\{\max\{\gamma_i', 0\}, 1\}$.
19: $\quad$ Determine number of samples to keep from $c_i$: $k_i = \text{round}(|c_i|(1 - \gamma_i'))$.
20: $\quad$ Select $k_i$ samples from $c_i$ (e.g., by re-applying density pruning within $c_i$ with an adjusted threshold $\tau_i$, or selecting $k_i$ least dense/prototype samples) to form $c_i^s$.
21: $\quad$ Add to final selected set: $D_s \leftarrow D_s \cup c_i^s$.
22: **return** $D_s$.

---

# E IMPLEMENTATION AND EXPERIMENT DETAILS

## E.1 DATASETS AND TRAINING PIPELINES

To thoroughly evaluate the efficacy of the proposed method, we perform extensive empirical analyses on three widely recognized standard benchmarks with practical and relevant object detection applications: **Waymo Open Dataset**, **COCO (Common Objects in Context) Dataset**, and the **nuScenes Dataset**.

**Waymo Open Dataset** The **Waymo Open Dataset (Waymo)** (Sun et al., 2020) is a comprehensive large-scale autonomous driving benchmark. We used version 1.3.1. Comprising 798 training sequences and 202 validation sequences, the data is originally captured at 10 Hz. Due to the high data volume and frame rate, we adopt the 2Hz subsampled version from Li et al. (2024), downsampling sequences by selecting every 5th frame. Additionally, bounding boxes that are not visible within any camera image views were filtered out from both training and validation splits. In line with established practice, our evaluations employ three standard detection categories: vehicles, pedestrians, and cyclists. We adopt mean Average Precision (mAP) at two distinct Intersection-over-Union (IoU) thresholds of 0.5 and 0.7 for performance measurement as commonly implemented for this dataset.

**COCO (Common Objects in Context) Dataset** The **COCO (Common Objects in Context) 2017** dataset (Lin et al., 2014) is a broadly recognized large-scale computer vision benchmark supporting object detection, instance segmentation, and captioning tasks. This work builds on the 2017 release, including approximately 118k training images with 886,284 annotated object instances spanning 80 diverse categories. These categories range from everyday objects such as vehicles and animals to more specialized entities including fashion accessories and sports gear. Following standard evaluation protocols, detection performance is measured using the Average Precision (AP) metric computed over IoU thresholds ranging from 0.50 to 0.95 with increments of 0.05, as defined by the primary COCO challenge metric (commonly termed as AP@[0.50:0.05:0.95]).

**nuScenes Dataset** The **nuScenes Dataset** (Caesar et al., 2020) constitutes 1,000 driving scenes, each around 20 seconds in duration, with key samples annotated at 2 Hz frequency intervals. Each annotated key frame contains RGB image inputs collected from six cameras, collectively providing full 360° horizontal field-of-view coverage. For the detection task considered here, annotations comprise 1.4 million 3D object bounding boxes spanning 10 different object classes: cars, trucks, buses, trailers, construction vehicles, pedestrians, motorcycles, bicycles, traffic cones, and barriers. Consistent with the standard nuScenes evaluation procedure, detection accuracy is evaluated through mean Average Precision (mAP), where matchings of predictions to ground-truth objects are determined based upon object-center distances projected onto the ground plane rather than by conventional 3D Intersection-over-Union (IoU).

**Model Training.** We follow established baseline models and training setups validated across the literature. For the Waymo and nuScenes datasets, we employ BEVFormer-S (a static variant of BEVFormer (Li et al., 2024)) with a ResNet101-DCN (He et al., 2016) backbone, initialized from FCOS3D proposals. Since data pruning concerns the utility of individual images, we adopt the static version BEVFormer-S, which drops the channel for temporal information between consecutive frames. For the COCO dataset, we deploy the well-established Faster R-CNN (Ren et al., 2016) framework with a ResNet-101-FPN backbone, implemented via Detectron2 (Wu et al., 2019).

All models are trained maintaining their original learning rate schedules and other hyperparameters detailed in their respective baseline implementations (e.g., batch size, optimizer choices). For Waymo and nuScenes datasets, training proceeds for 12 epochs on both full and pruned datasets, employing batch sizes of 8 scenes per iteration. The number of training epochs and the learning rate schedule remain unchanged when using pruned data subsets. For the COCO dataset, we follow the standard 270K-iteration training scheme for the full dataset, adopting batch sizes of 16 images per step, which effectively yields about 36.6 epochs over the 118k-image training set. When using pruned subsets of COCO, the number of training iterations is scaled linearly in proportion to the size of the reduced training subset, thus maintaining a consistent effective number of epochs. Correspondingly, step-based learning rate schedules for COCO training are also proportionally scaled.

All model training was conducted on single nodes equipped with 8x NVIDIA V100 16GB GPUs. Each training run typically took around one day. Each experiment has been repeated over 3 runs, with standard deviations reported alongside the main results. The computation overhead of each training run is directly proportional to the size of the respective training subset, and efficiency improvements from data pruning are materialized as savings in computational expense for model training. VLM captioning, as part of the dataset preprocessing, was conducted on a single NVIDIA A5880 Ada 48GB GPU.

We refer to the original papers of the baseline models for further details on dataset preprocessing, specific model configurations, and training hyperparameters.

### E.2 BASELINES

We benchmark our proposed method, **ADADEDUP**, against three representative baseline approaches: **Random Downsampling (Random)**, **Visual Deduplication (CLIP-DeDup)**, and **VLM-SSE (SSE)**. These are selected to cover a diverse range of practical and relevant data pruning setups.

**Random Downsampling (Random)**  This is a straightforward yet widely adopted baseline, prevailing in large-scale data pipelines due to its simplicity and computational efficiency. Specifically, this approach uniformly samples a subset of data from the original dataset without replacement. Random downsampling is commonly utilized, and sometimes essential, when dealing with excessively large datasets that exceed computational resources.

**Visual Deduplication (CLIP-DeDup)**  This baseline represents a density-based pruning strategy. Data samples are first embedded into a learned visual embedding space. Samples are discarded if their visual embeddings are within a specified proximity threshold of another sample's embeddings, aiming to reduce visual redundancy to achieve the target dataset size. For the Waymo and nuScenes datasets, we use the pretrained CLIP-ViT-L/14 model (Radford et al., 2021) to extract visual embeddings. For the COCO dataset, we utilize the GroundingDINO model (Liu et al., 2024b) (features aggregated from ground-truth object queries). Specifically for COCO, the GroundingDINO model uses a Swin-L backbone; ground-truth object labels serve as input queries, and output features are aggregated via max-pooling to form compact visual representations. The Waymo and nuScenes datasets are structured by scenes, each comprising multiple images captured simultaneously from different viewpoints. For these datasets, visual deduplication is conducted using only the front-view images; if a front-view image is flagged as redundant and removed, all correlated images within that scene are consequently discarded. For the COCO dataset, deduplication is performed directly on each individual image independently.

**VLM-SSE (SSE)**  This method, **VLM-SSE (SSE)** (Shen et al., 2025), is a state-of-the-art semantic deduplication technique. It enhances visual deduplication by first generating image captions using a Vision-Language Model (VLM) to form semantic clusters. Within each cluster, visual deduplication is then performed using a uniform threshold. This approach, related to methods like SemDeDup (Abbas et al., 2023), aims to ensure images are eliminated only when they are both semantically and visually similar. Image captions are generated using the pretrained LLaVA-1.5-13B VLM (Liu et al., 2024a). The generated textual descriptions group images into semantic clusters. We test with 10, 100, and 500 clusters to evaluate performance under different semantic granularities. Similar to the CLIP-DeDup baseline, for the Waymo and nuScenes datasets, VLM captioning and subsequent cluster assignment occur exclusively based on the front-view images, with pruning decisions affecting the entire scene. For COCO, captions are generated and deduplication is applied individually for all images. *The full prompts used for all datasets and samples for generated captions are provided in Appendix F.1.*

## F ADDITIONAL RESULTS AND VISUALIZATIONS

### F.1 VLM PROMPTS AND SAMPLE GENERATED CAPTIONS

Prompts used to generate VLM captions for each dataset and samples for generated captions are provided in boxes D.1.1 and D.1.2. For AV datasets, Waymo and nuScenes, we used the Specialized

AV Prompt from Shen et al. (2025). For the generic object-detection dataset, COCO, we developed the original prompt following the same methodology. The dense captions well characterize the semantic content of the images while emphasizing on content of interest.

---

### D.1.1 Specialized AV Prompt for AV Datasets (Waymo, nuScenes)

- **Specialized AV Prompt (Shen et al., 2025):** The image is taken from inside the ego vehicle looking out through the windshield onto a road and you are the driver of the ego vehicle. Please describe the driving condition including the location, weather, road users, and their motions. During your description, there are several things to keep in mind. 1. Please pay attention only to the objects on the driving roads and ignore the background. 2. Ignore the brands of the vehicles. 3. Describe it if objects are partially occluded by others, or are in areas with different brightness such as under shades. Please provide a concise description in one paragraph with less than 150 words.

- **Input Image (Waymo):**

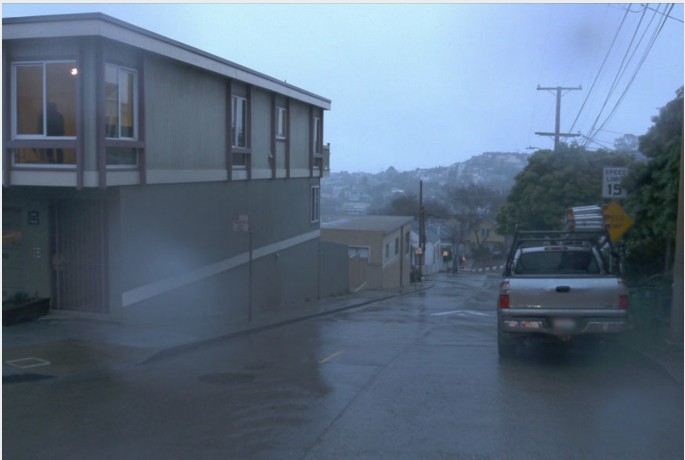

- **VLM Generated Caption:** The image depicts a driving scene on a wet road, likely due to recent rain. The road is lined with parked cars on the right side, and a truck is visible in the distance. The road curves slightly to the right, and there is a speed limit sign indicating 15 mph. The weather appears overcast, and the lighting is somewhat dim, suggesting it might be early morning or late afternoon. The road surface is slick, and the visibility is somewhat reduced due to the weather conditions. The driver of the ego vehicle should exercise caution and maintain a safe speed to avoid accidents.

### D.1.2 Generic Prompt for COCO Dataset

- **Generic Prompt:** The image is a common photo. Please describe the detailed content of the image including the context, scenario, main objects as well as other objects (be comprehensive). Also, describe the style of the photo, shooting perspective, lighting condition, distance, and the relative size of the objects in the image. Please provide a concise description in one paragraph with less than 150 words.

- **Input Image (COCO):**

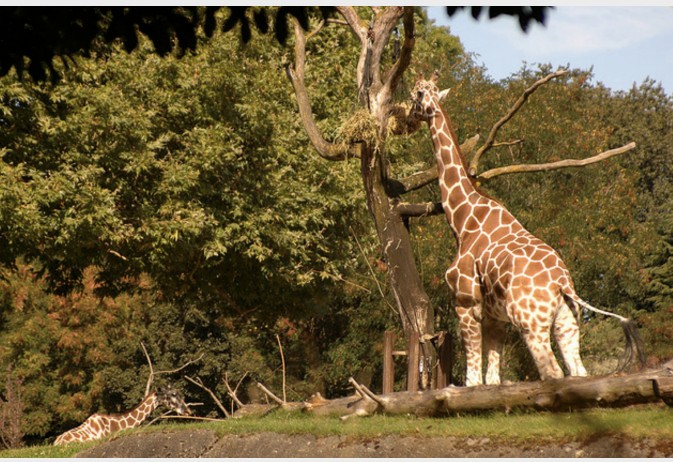

- **VLM Generated Caption:** The image captures a serene moment in a zoo enclosure where two giraffes are engaged in feeding. The giraffe in the foreground, with its long neck extended, is reaching for a branch laden with hay. Its companion stands a bit further back, also partaking in the meal. The enclosure is a lush greenery of trees and bushes, providing a naturalistic habitat for the animals. The photo is taken from a low angle, emphasizing the towering height of the giraffes and the verdant surroundings. The lighting is bright, suggesting it's daytime, and the overall composition conveys a sense of tranquility and harmony with nature.

### F.2 VISUALIZING RETAINED AND PRUNED EXAMPLES FOR EACH SEMANTIC CLUSTER OF WAYMO DATASET

Illustration on density-based data pruning (Shen et al., 2025) with 10 clusters on Waymo Dataset, retaining 60% of images (as in Figures 1 and 2).

Figure 6 shows visualizations of representative samples from 10 semantic clusters derived from the Waymo dataset via clustering on caption embeddings. Notable findings include: (i) Distinct semantic themes are evident within each cluster, for instance, nocturnal scenes in Cluster #4 and adverse weather conditions (fog/rain) in Cluster #9. (ii) Samples pruned for intra-cluster visual redundancy often represent static scenarios, particularly stopped traffic (Clusters #2, #7, #9, characterized by red traffic or brake lights), leading to high frame-to-frame similarity. (iii) A significant portion of other pruned images corresponds to stable environments (e.g., Clusters #1, #3, #5, depicting tranquil residential areas) where the visual perspective remains relatively constant despite ego-vehicle motion.

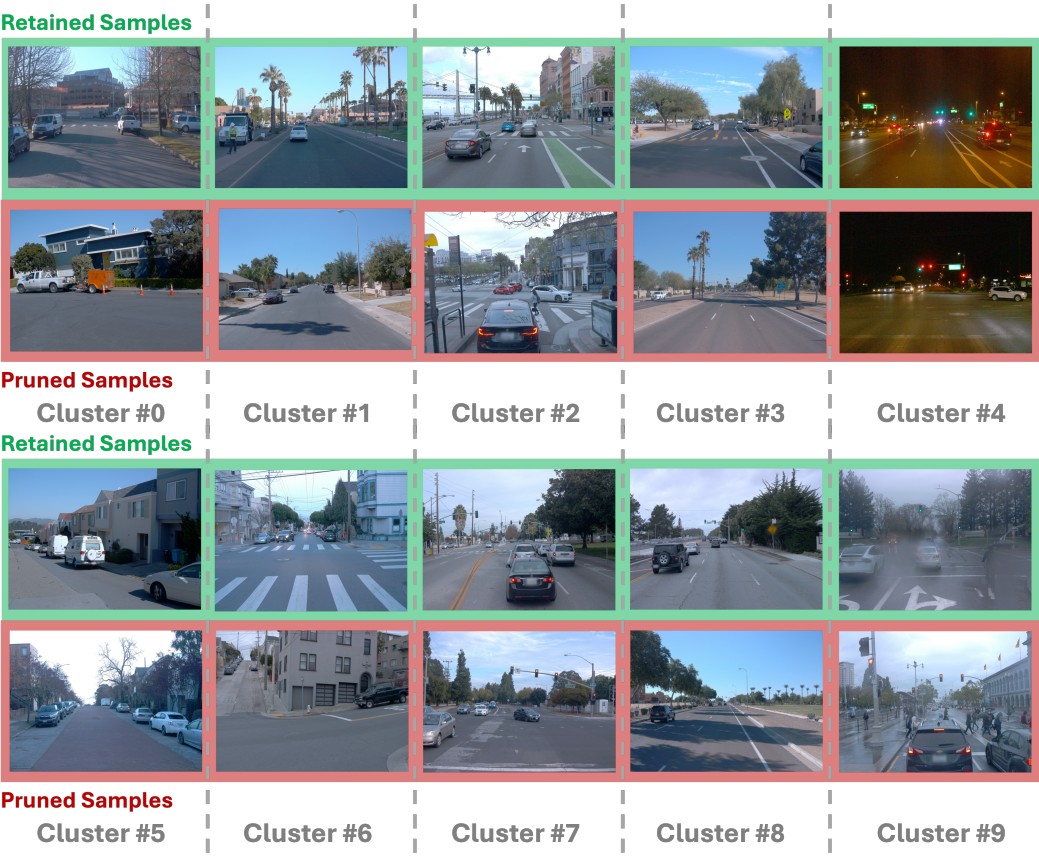

Figure 6: Illustration on density-based data pruning (Shen et al., 2025) with 10 clusters on Waymo Dataset, retaining 60% of images (as in Figures 1 and 2). Example images from 10 semantic clusters of the Waymo dataset, derived from caption embedding clustering. Each cluster displays a distinct visual theme (e.g., night scenes in Cluster #4, foggy/rainy conditions in Cluster #9). Visually redundant samples were pruned, many of which depicted stopped traffic (Clusters #2, #7, #9, indicated by red traffic or brake lights) or static scenes with fixed perspectives despite ego-vehicle motion (Clusters #1, #3, #5, e.g., tranquil residential areas).

