# OpenReview forum: "AdaDeDup: Adaptive Hybrid Data Pruning for Efficient Object Detection Training"
_ICLR.cc/2026/Conference — ICLR 2026 Conference Withdrawn Submission_

### Official Review · Reviewer_WQz7 · 2025-10-26

**Soundness:** 2
**Presentation:** 3
**Contribution:** 2
**Rating:** 2
**Confidence:** 4

**Summary:**

The authors introduce ADADEDUP, a data pruning framework designed for object detection that combines density-based pruning (removing redundant samples based on VLM embedding similarity) with model-informed adaptive feedback. Samples from the datasets are first clustered with K-Means. De-duplication is then performed within each cluster, which results in an initial selected subset. A proxy model is trained on this set, and a differential loss signal is computed for each cluster to determine its relative information content and guide the adjustment of the pruning strength for each cluster. The authors evaluate the method on Waymo, COCO, and nuScenes using BEVFormer and Faster R-CNN, and they choose Random Downsampling, CLIP-DeDup, and VLM-SSE as baselines. Results show consistent improvements over baselines at different pruning ratios. The method also achieves near-original performance at 20% pruning on Waymo and 15% on COCO.

**Strengths:**

- **Originality:** The paper tackles a somewhat underexplored problem of data pruning for large-scale object detection. The idea of using proxy-model loss differentials to adjust pruning intensity per cluster represents an extension of prior deduplication and coreset ideas.

- **Quality and clarity:** The paper is well written and presented, with figures and illustrations that help illustrate the behavior of the method.

- **Significance:** Its potential impact lies in improving dataset curation and reducing computational overhead for object detection pipelines. It is of note that the method achieves near-original model performance while pruning 15-20% of the original data. The method also appears computationally cheaper than other model-based pruning methods.

**Weaknesses:**

- Lack of deep analysis: analysis experiments are completely lacking, e.g., ablations on method components, sensitivity to the embedding extraction method (different VLM models vs pre-trained vision models), analysis of the impact of the proxy model used, and sensitivity to certain hyperparameters such as β, α+, α-, etc.

- The authors claim that the method is “dramatically cheaper than model-based pruning methods”, though this claim of efficiency has not been quantified. Extracting pre-trained embeddings (VLM features) and proxy training still implies some nontrivial compute. Providing wall-clock comparisons, for example, would substantiate the claim made by the authors.

- The empirical validation is limited to Faster R-CNN for COCO and BEVFormer-S for Waymo and nuScenes. This narrow evaluation fails to test the method’s robustness across several architectures in both homogeneous and heterogeneous settings. A strong validation would have been: prune data using a proxy trained with one architecture (e.g., Faster R-CNN), then train different models (e.g., Faster R-CNN, YOLO series, DETR,..) on that pruned set. Because AdaDeDup uses model loss from a proxy network, its success could depend on the specific architecture’s loss landscape or sensitivity to sample difficulty. Thus, experiments need to be carried out to evaluate the method’s cross-architecture generalization.

**Questions:**

- Can the authors clarify the specific initial pruning strategy used? Is the density-based pruning applied globally or within clusters? Is a distance threshold used to de-duplicate samples within each cluster, or are prototypes selected from each cluster based on some metric? This should be clarified in Algorithm 1 and Section 3.3.1.
- Can the authors provide some, if not all, of the ablation studies suggested in the weaknesses section?
- Can the authors quantify the claim that their method is dramatically cheaper than previous methods?
- Have the authors examined whether AdaDeDup biases class or scene distributions? A per-class pruning ratio could clarify this.

---

### Official Review · Reviewer_zS8f · 2025-10-31

**Soundness:** 4
**Presentation:** 3
**Contribution:** 3
**Rating:** 4
**Confidence:** 4

**Summary:**

This paper proposes ADADEDUP, a novel two-stage hybrid data pruning framework designed to improve the efficiency of training large-scale object detection models. The method addresses the limitations of purely density-based methods (which can be task-agnostic) and model-based methods (which are computationally expensive). ADADEDUP first performs an initial, efficient density-based pruning on semantically clustered data. The core innovation is the second stage: it trains a lightweight "proxy" model on this initially pruned subset and then uses it to evaluate the loss on both the kept and pruned samples within each cluster. By comparing these losses, it creates a "differential loss" signal that identifies clusters where informative data was likely discarded. This signal is used to adaptively adjust the pruning thresholds for each cluster—pruning more from redundant clusters and less from informative ones—before producing the final data subset. Extensive experiments on Waymo, COCO, and nuScenes show that ADADEDUP significantly outperforms strong baselines, achieving performance nearly identical to training on the full dataset while pruning up to 20% of the data.

**Strengths:**

1. Intuitive and Well-Motivated Method: The core idea of using a proxy model's loss to adaptively refine an initial density-based pruning is clever and directly addresses a known weakness of task-agnostic pruning. The cluster-adaptive mechanism provides a principled way to balance redundancy removal and information preservation without incurring the prohibitive cost of full-scale model-based methods. The motivation and mechanism are exceptionally well-illustrated in Figure 1.
2. Comprehensive and Strong Empirical Evaluation: The paper is validated with an extensive set of experiments (>50k GPU hours) on three challenging, large-scale object detection benchmarks (Waymo, COCO, nuScenes). The authors compare against well-chosen and strong baselines, including random sampling, a standard density-based method (CLIP-DeDup), and a state-of-the-art semantic deduplication method (VLM-SSE). The results are consistently strong across all datasets and pruning ratios, clearly demonstrating the effectiveness of the proposed adaptive strategy.

**Weaknesses:**

1. Novelty:
The paper's novelty lies in the specific mechanism of cluster-adaptive re-pruning, not in the foundational concepts. The use of hybrid methods, clustering before pruning (as in VLM-SSE), and leveraging proxy models (as in Coleman et al., 2019) are established ideas. The contribution is a smart and effective synthesis and refinement of these existing components. While highly effective, it could be viewed by some as an incremental (though powerful) improvement over prior art rather than a completely new paradigm.

2. Experiments:
- Proxy Model Sensitivity: The performance of ADADEDUP relies on the "differential loss" signal from a proxy model. However, the paper does not sufficiently analyze how the quality of this signal depends on the proxy model itself. The choice of proxy architecture, its size, and its training budget (e.g., trained on "<=10k samples") are critical parameters, but their impact on the final pruning effectiveness is not explored. A weak proxy might provide a noisy signal, potentially degrading the adaptive pruning.
- Computational Overhead Analysis: The paper claims to be more efficient than model-based methods, which is plausible. However, it lacks a direct comparison of the computational cost of the pruning process itself. The proposed method requires feature extraction, clustering, initial pruning, training a proxy model, and a full inference pass to calculate losses, which is significantly more complex than the baselines. A table detailing the wall-clock time or total FLOPs for ADADEDUP vs. VLM-SSE to generate the final pruned dataset would make the efficiency claims more concrete.
- Hyperparameter Sensitivity: The method introduces several key hyperparameters, including the number of clusters (K), the adaptation strength (β), and the loss scaling factors (α+, α-). While the paper provides some heuristics for setting β, a more thorough sensitivity analysis for these parameters, especially for K, is missing. Baseline performance (VLM-SSE) is shown to vary with K, and it is likely ADADEDUP's performance does as well.
- Additional Things:
1. Hard Samples vs. Outliers: A significant weakness, which the authors briefly acknowledge in the conclusion, is the inability to distinguish valuable "hard" samples from noisy or mislabeled "outliers." Both would likely exhibit high loss, and the current method risks preferentially keeping noisy data. This is a classic challenge for loss-based data selection methods that is not addressed here.
2. Looseness of Theoretical Connection: Section 3.2 attempts to frame the differential loss heuristic as a "zero-order estimation for the policy gradient" of a bi-level optimization problem. This connection feels somewhat post-hoc and tenuous. The method presented in Algorithm 1 is a practical and effective heuristic, but the justification via policy gradients may overstate the theoretical grounding, as the proxy for the gradient (the difference in loss on kept vs. pruned samples) is quite coarse.
3. Additional baseline and related works: the data pruning topic is aligned with the goal of reducing the data amount to enhance the effieicny. It is better to compare with data distillation works (or at least cite and discuss them properly) [1].

[1] Wang S, Yang Y, Liu Z, et al. Dataset distillation with neural characteristic function: A minmax perspective[C]//Proceedings of the Computer Vision and Pattern Recognition Conference. 2025: 25570-25580.

**Questions:**

Please see Weaknesses.

I could consider raising my scores if additional experiments and discussion are added into the new version of paper.

---

### Official Review · Reviewer_KawY · 2025-11-01

**Soundness:** 2
**Presentation:** 2
**Contribution:** 2
**Rating:** 2
**Confidence:** 5

**Summary:**

Data pruning is a promising solution to address the challenges posed by the computational burden and inherent redundancy of large-scale datasets in training contemporary machine learning models. This paper introduces a hybrid data pruning framework that synergistically integrates density-based clustering with model-informed, cluster-adaptive feedback to enhance data efficiency in training large-scale models, particularly for object detection.

**Strengths:**

1. The computational burden and inherent redundancy of large-scale datasets is a noteworthy issue, and this paper explores this problem.
2. This paper integrates previous density-based approaches with model-based techniques, demonstrating certain advantages.

**Weaknesses:**

1. Lack of innovation. This paper can be seen as a combinatorial optimization of previous density-based and model-based methods, but it lacks substantial originality.
2. The experimental section only utilized outdated BEVFormer and Faster R-CNN detectors, lacking validation of effectiveness on the latest state-of-the-art detectors.
3. The proposed method relies on pre-trained VLMs, raising the issue of unfair comparison.
4. The proposed method has only been validated on a single detection task and lacks effectiveness verification for extension to other visual tasks.
5. The proposed method still requires substantial computational resources.

**Questions:**

1. Please analyze the impact of the number of clusters.
2. How to ensure the reliability of the initial model?

---

### Official Review · Reviewer_UbpY · 2025-11-01

**Soundness:** 3
**Presentation:** 3
**Contribution:** 3
**Rating:** 6
**Confidence:** 2

**Summary:**

This paper proposes AdaDeDup (Adaptive Hybrid Data Deduplication), a hybrid framework for large-scale visual dataset pruning that balances efficiency and task relevance. The authors observe that density-based deduplication is efficient but task-agnostic, while model-based selection is task-aware but computationally expensive. AdaDeDup addresses this gap by first performing density pruning within semantic clusters using VLM embeddings, then employing a lightweight proxy model to compare the loss between retained and discarded samples in each cluster, adaptively adjusting the pruning ratio per cluster.

Experiments on Waymo, COCO, and nuScenes demonstrate that AdaDeDup consistently outperforms baselines such as CLIP-DeDup and VLM-SSE under the same pruning rates—for instance, reducing over 50% of the performance loss compared to random sampling on Waymo, and maintaining near-full performance on COCO even after pruning 20% of the data. Overall, the paper presents a well-motivated and practical approach to cluster-level adaptive data deduplication with solid empirical validation.

**Strengths:**

1. The paper clearly identifies the gap between efficiency-oriented density-based deduplication and task-aware yet costly model-based approaches, then proposes a hybrid and cluster-adaptive framework to bridge them. The motivation, design logic, and visual explanations (e.g., Fig. 1–2) are coherent and well-connected, making the problem statement convincing and well-motivated.
2. Experiments across three major detection datasets—Waymo, COCO, and nuScenes—consistently demonstrate that AdaDeDup outperforms representative baselines under the same pruning ratios. The improvements are stable, and the method retains near-full performance under moderate pruning (20–30%), confirming its practical effectiveness.
3. The cluster-level loss-difference signal offers an intuitive and explainable mechanism for adaptive threshold adjustment. This design maintains the scalability of density-based pruning while incorporating task relevance through proxy-model feedback, striking a good balance between performance and simplicity.
4. The paper provides detailed hardware settings, training budgets (>50 k GPU hours), and multiple runs with standard deviations, indicating strong experimental rigor and reproducibility. The framework is lightweight enough to integrate into large-scale detection pipelines, showing clear potential for practical deployment.

**Weaknesses:**

1. The method depends on several preset parameters — such as the number of clusters $K$, adaptation strength $\beta$, and scaling factors $\alpha_{+}$ and $\alpha_{-}$ — yet the paper provides no ablation or sensitivity experiments. These parameters may behave differently across datasets, raising concerns about generalization and robustness.
2. The adaptive adjustment depends on a proxy model to estimate the loss difference $\Delta \ell_i$ between retained and pruned samples. However, the paper does not analyze how the proxy model’s capacity, architecture, or training data affect the $\Delta \ell_i$ signal, leaving uncertainty about its stability and reliability.
3. For multi-camera datasets (Waymo, nuScenes), pruning is determined solely by the front-view image, and all other camera views of the same scene are removed simultaneously. While this simplifies the pipeline, it risks discarding complementary information from other perspectives, potentially limiting performance in multi-view detection settings.

**Questions:**

The paper highlights that density-based deduplication is faster while model-based methods are slower, and the proposed approach combines both as a hybrid. To substantiate the claimed balance between efficiency and effectiveness, could the authors include an efficiency comparison during the data pruning stage?

---

### Official Review · Reviewer_4J4Q · 2025-11-02

**Soundness:** 3
**Presentation:** 2
**Contribution:** 3
**Rating:** 4
**Confidence:** 3

**Summary:**

This paper introduces AdaDeDup, an adaptive hybrid data pruning framework designed to improve data efficiency in large-scale object detection training. The method combines density-based pruning with task-aware feedback from a proxy model at the cluster level—first partitioning data semantically, pruning based on density, and then further adapting pruning ratios within clusters using model loss signals.

**Strengths:**

This paper addresses the highly relevant and practical problem of data pruning for large-scale object detection. The core innovation lies in the two-stage, adaptive process, combining previous density-based and Model-Based pruning methods.

**Weaknesses:**

The paper has an unclear organizational structure, such as subsections "The Challenge of Optimal Selection Necessitates Approximate Solutions" L224 and "Limitations of Fixed Density-based Pruning" L246, which read more like extensions of the "Related Work" section or as general motivation.

The entire adaptive stage (Stage 2) is dependent on a proxy model $\tilde{\mathcal{A}}$ that is trained only on the initially pruned subset $D_s^{(0)}$. This introduces a significant risk of confirmation bias: if the initial density-based pruning (Stage 1) erroneously discards a cluster of challenging-but-informative samples, the proxy model $\tilde{\mathcal{A}}$ will never see them during training. Consequently, when $\tilde{\mathcal{A}}$ evaluates these pruned samples, they will naturally exhibit a high loss ($l_i^p$) merely because they are out-of-distribution for the model, not necessarily because they are informatively valuable. This could mislead the $\Delta l_i$ signal, causing the framework to reinforce, rather than correct, the initial pruning's flaws.

While the method is presented as efficient, it glosses over the non-trivial computational overhead. It requires (1) fully training a proxy model on a large subset, and (2) performing a full inference pass the entire original dataset $D_a$ (both $D_s^{(0)}$ and $D_p^{(0)}$) over a 13B LLaVA model. This extra cost is not quantified, and the paper lacks a cost-benefit analysis to justify the marginal gains.

**Questions:**

See "Weaknesses".

---

### Note · Authors · 2025-11-14

I have read and agree with the venue's withdrawal policy on behalf of myself and my co-authors.